# Comparative Study of the Co-Occurring *Alternaria* and *Colletotrichum* Species in the Production of Citrus Leaf Spot

**DOI:** 10.3390/jof9111089

**Published:** 2023-11-08

**Authors:** Mengying Lei, Congyi Zhu, Luoye Li, Jiangshan Liu, Jiashang Liu, Feng Huang

**Affiliations:** 1College of Forestry Engineering, Guangdong Eco-Engineering Polytechnic, Guangzhou 510520, China; mylei88@126.com (M.L.); liluoye822@163.com (L.L.); ljs20202023@126.com (J.L.); berry1176@126.com (J.L.); 2Plant Protection Research Institute, Guangdong Academy of Agricultural Sciences, Key Laboratory of Green Prevention and Control on Fruits and Vegetables in South China Ministry of Agriculture and Rural Affairs, Guangdong Provincial Key Laboratory of High Technology for Plant Protection, Guangzhou 510640, China; 3Key Laboratory of South Subtropical Fruit Biology and Genetic Resource Utilization (MOA) & Guangdong Province Key Laboratory of Tropical and Subtropical Fruit Tree Research, Institute of Fruit Tree Research, Guangdong Academy of Agricultural Sciences, Guangzhou 510640, China; zhucongyi@hotmail.com

**Keywords:** Alternaria brown spot, anthracnose, citrus, *Colletotrichum*, transcriptional sequencing

## Abstract

Both of the two citrus diseases, Alternaria brown spot (ABS) and Anthracnose, caused by *Alternaria* and *Colletotrichum* spp., respectively, can produce leaf lesions which are hard to differentiate. These two diseases have been confused as causal agents of brown spot for over a decade in China. In this study, citrus leaves with or without brown spot were collected from Zhaoqing, Guangdong and Wanzhou, Chongqing, and were further used for the taxonomic and functional comparisons between the co-occurring *Alternaria* and *Colletotrichum* species. In the amplicon sequencing, the average relative abundance and the composition of *Alternaria*, but not *Colletotrichum*, increased (from 0.1 to 9.9, *p* = 0.059; and to 0.7, *p* < 0.05) and significantly altered (*p* < 0.01) with the brown spot in Zhaoqing and Wanzhou, respectively. Two representative isolates *Alternaria* sp. F12A and *Colletotrichum* sp. F12C, from the same brown spot, were proved with different virulence and host response activation to citrus leaves. F12A caused typical symptoms of brown spot with the average spot length expanded to 5 and 6.1 cm, and also altered the citrus global gene expression 48 and 72 h after inoculation. In addition, F12A enriched the expression of genes that were most frequently involved in plant defense. In comparison, F12C caused leaf spot limited to the wounded site, and its milder activation of host response recovered 72 h after inoculation. Our study indicates that the incidence of brown spot in China is caused by *Alternaria* species, and the ABS should be a fungal disease of major concern on citrus.

## 1. Introduction

Citrus is broadly cultivated in many different countries around the world [1]. During the growth of citrus, many plant diseases, caused by microbial pathogens, can occur on the above- and underground organs of citrus and cause huge economic losses to the citrus industry [2,3]. Among the well-known citrus microbial diseases, Alternaria brown spot (ABS), caused by *Alternaria* spp., and Anthracnose, caused by *Colletotrichum* spp., are two of the most important foliar fungal diseases [2,3]. The typical leaf symptoms of ABS are brown to black lesions, surrounded by a yellow halo, which can later lead to the dying back of twigs [4,5]. Similarly, Anthracnose also causes leaf lesions, and leads to “withertip”, namely, the dying back of twigs [2,3]. In addition, *Colletotrichum gloeosporioides* Penz., the dominant causal agent of citrus Anthracnose, has world-wide distribution, and is found universally as endophytes and saprophytes on both healthy and diseased citrus trees [3,6]. This means that Anthracnose occurs along with the wounded and weakened leaves; those leaves, infected by ABS, may be colonized by *Colletotrichum gloeosporioides* as a secondary pathogen and a saprophyte [3,7].

The confusion of ABS and Anthracnose, or *Alternaria* species and *Colletotrichum* species, for the symptoms of citrus brown spot has lasted for over a decade in China, but still lacks a conclusion [8,9]. ABS was first observed in 2003, and reported in 2010, in Yunnan Province, China [10]. After that, the disease has been successively reported as an outbreak in several other citrus production areas in China, such as Chongqing [7], Zhejiang [11], Guangxi [9], and Fujian [12]. However, the diseased citrus trees (*C. reticulata* var. *Gonggan*, a highly sensitive variety to ABS), with similar ABS symptoms in Zhaoqing, Guangdong, were identified as a result of the outbreak of citrus Anthracnose in 2008 and 2010 [8,13]. From then on, citrus Anthracnose has been a major destructive fungal disease in this citrus production area, with different phytopathologists screening and recommending fungicides used to control *Colletotrichum gloeosporioides*, to control it [14,15,16]. Considering *Colletotrichum gloeosporioides*, which widely exists with citrus in China [17,18], this raises a question why an endophyte, saprophyte and latent pathogen has been prevalent in this area as a destructive pathogen for such a long time.

For the last three years, citrus trees with brown spot (Figure 1A–C) have been very common in the orchards in Zhaoqing, Guangdong. Citrus leaves were largely fallen (Figure 1D,E) and economic losses were caused by the outbreak of this disease from February to April each year. In one orchard of Guanxu, Deqing, Zhaoqing, the orchardists found that their regular control methods for citrus Anthracnose could not effectively control the brown spot disease anymore. Based on these, citrus leaves with or without brown spot (Figure 1A,B) were collected from Zhaoqing, Guangdong and Wanzhou, Chongqing, and subjected to culture-dependent and -independent studies. From these, we aimed to compare the occurrence of and variation in *Alternaria* and *Colletotrichum* species on citrus leaves due to brown spot and compare the virulence of the representative co-occurring *Alternaria* and *Colletotrichum* species to citrus, and in accordance, the strength of the host response to the two species.

## 2. Materials and Methods

### 2.1. Leaf Collection

The citrus leaves, without any symptoms (asymptomatic, Asym) or with Alternaria brown spot (ABS, Figure 1A,B), were collected in three orchards in Guanxu, Zhaoqing City of Guangdong Province and Xiangshui, Wanzhou District of Chongqing Municipality, respectively (Table 1). The varieties were *Citrus reticulata* cv. *Gonggan* and *C. reticulata* cv. *Hongjv* in Zhaoqing and Wanzhou, respectively. In each orchard, 10 to 15 diseased trees were selected for sampling. From each tree, one Asym leaf and one ABS leaf, about the same age, were picked and stored in sterile plastic bags. The leaves were put on ice and sent to the lab as soon as possible. All leaves were surface-sterilized by soaking in 1 × TE buffer (supplemented with 0.1% Triton X-100) for 30 s, 75% ethanol for 15 s, 2% bleach for 15 s, and then rinsed in sterile water for three times to remove leaf epiphytic microbes [19].

### 2.2. Fungal Isolation and Identification

The leaf segments (3 × 3 mm^2^), including one third of necrotic tissue and two thirds of healthy tissue, were cut from the edge of the brown spot. Then, the leaf segments were placed on potato dextrose agar (PDA: potato extract 4 g, dextrose 20 g, agar 15 g per 1 L water) plates supplemented with or without 10 μg/mL carbendazol to inhibit the growth of *Colletotrichum* [7]. All plates were incubated under 25 °C, 12/12 light/dark cycles for seven days. The emerged fungal isolates were purified by moving the mycelia (3 mm in diameter) from the colonial margin to a new PDA plate. The new plates were incubated under the same condition for two weeks, and then, according to the morphology of their conidia, the isolates were identified to the genus level (*Alternaria*, *Colletotrichum*, or others).

### 2.3. DNA Extraction and Amplicon Sequencing

The leaves were ground with liquid nitrogen and the microbial DNA was extracted from approximately 1 g leaf powder using the cetyltrimethylammonium bromide method [20]. The DNA concentration and purity was assessed by electrophoresis in 1% agarose gel and then the DNA solution was diluted to 1 ng/µL with sterile water. Partial nucleotide sequence of the fungal nuclear ribosomal internal transcribed spacer region (ITS rDNA) was amplified by PCR with the fungal primer set ITS1-1F (5′-CTTGGTCATTTAGAGGAAGTAA-3′) and ITS1-1R (5′-GCTGCGTTCTTCATCGATGC-3′) for each sample [21]. The PCR amplification was conducted as follows: 15 µL of the Phusion^®^ High-Fidelity PCR Master Mix (New England Biolabs, Ipswich, MA, USA), 2 µM of the forward and reverse primers, and approximately 10 ng template DNA. The thermal cycling program was set as follows: initial denaturation at 98 °C for 1 min, followed by 30 cycles of denaturation at 98 °C for 10 s, primer annealing at 50 °C for 30 s, and extension at 72 °C for 30 s, and a final extension at 72 °C for 5 min. For each leaf sample, the PCR amplification was conducted in triplicate and the final PCR products were pooled to form one sample for later use. The pooled PCR product was electrophoresed in 2% agarose gel, and the target DNA band was cut and purified with the Qiagen Gel Extraction Kit (Qiagen, Hilden, Germany). The DNA library for sequencing was generated using the TruSeq^®^ DNA PCR-Free Sample Preparation Kit (Illumina, San Diego, CA, USA) and amplicon-sequenced on the Illumina NovaSeq platform by the Novogene Bioinformatics Technology Co., Ltd. (Tianjin, China).

After sequencing, raw reads were filtered using Trimmomatic v0.39 [22]. The primer sequences were removed using Cutadapt v1.9.1 [23]. High-quality reads were assembled using FLASH v1.2.7 (http://ccb.jhu.edu/software/FLASH/ (accessed on 7 June 2022)). Then, denoising and removal of chimeric sequences were processed using the DADA2 plugin [24] adapted to the QIIME2 2020.6 workflow [25]. The amplicon sequence variants (ASVs) were clustered using DADA2, and annotated following Bayesian classification using UNITE (https://unite.ut.ee/ (accessed on 7 June 2022)) database.

### 2.4. Fungal Inoculation

Two co-occurring fungal isolates, *Alternaria* sp. F12A and *Colletotrichum* sp. F12C, were isolated from the same brown spot, and selected for inoculation. The isolates were grown on PDA plates under 25 °C, 12/12 light/dark cycles for seven days. To prepare the inoculants, 5-mm PDA discs, with or without fungal mycelia, were punched from the fungal colonial margin and plain PDA plate as control, respectively. The Gonggan leaves, about two thirds of the area of mature leaf, were picked and washed in tap water, and then rinsed in 2% bleach for 5 min. To facilitate the infection of *Colletotrichum* species, the leaves were scratched by a bunch of three sterile toothpicks twice at the upper right side of the midrib. A fungal disc or plain PDA disc was put upside down on the leaf-scratched site, all the leaves were then moved into white rectangular basins (42 × 33 × 12.5 cm^3^) and stored at room temperature for two or three days. The severity of necrotic spot was measured by the spot length along the leaf midrib.

### 2.5. RNA Extraction and Transcriptional Sequencing

In total, twenty-four leaf samples, inoculated with *Alternaria* sp. F12A, *Colletotrichum* sp. F12C, and plain PDA disc for 48 and 72 h, were collected. The leaf RNA was extracted using RNAprep Pure Plant Plus Kit (Tiangen Biotech Co., Ltd., Beijing, China), the concentration, purity, and integrity of the RNA and RNA library were examined using NanoDrop 2000 (Thermo Fisher Scientific, Wilmington, DE, USA) and the RNA Nano 6000 Assay Kit of the Agilent Bioanalyzer 2100 system (Agilent Technologies, Santa Clara, CA, USA). The RNA library was constructed using NEBNext UltraTM RNA Library Prep Kit for Illumina (NEB, Ipswich, MA, USA) following the manufacturer’s recommendations. The library preparations were then sequenced on an Illumina platform (San Diego, CA, USA) and paired-end reads were generated. The raw data were processed by removing reads containing adapter, reads containing ploy-N, and low quality reads to obtain clean data (178.72 Gb, at least 7.1 Gb for each sample). The high-quality clean data, with a minimum 94.6% quality score of Q30, were mapped to the genome of *C. reticulata* (http://citrus.hzau.edu.cn/orange/download/index.php (accessed on 6 July 2023)) with mapping ratios ranging from 90.4% to 94.6% (Appendix A). The genes were further used for the annotation of function against the databases of NCBI non-redundant protein sequences (Nr), clusters of orthologous groups of proteins (KOG/COG), KEGG Ortholog database (KO), Gene Ontology (GO), and also for the enrichment analysis of genes (GO) and pathways (KEGG). The gene expression level was estimated by the fragments per kilobase of transcript per million fragments mapped (FPKM) and the differential expression analysis was performed using DESeq2 [26].

### 2.6. Statistical Analysis

For amplicon sequencing, an ASV table of all samples was generated with QIIME2 2020.6 [25]. The ASVs assigned to *Alternaria* and *Colletotrichum* were extracted to calculate their ASV number and relative abundance. Permutational multivariate analysis of variance was used to assess the effects of disease (Asym and ABS) on the composition and structure of the *Alternaria* and *Colletotrichum* communities using the adonis command with 999 repetitions implemented in the vegan package [27]. For visualization, the Bray–Curtis distance matrix was subjected to principal coordinate analysis (PCoA) using the pcoa command in the Ape package [28] and plotted using the ggplot2 package [29]. For transcriptional sequencing, the resulting *p* values of differentially expressed genes (DEGs) were adjusted using Benjamini and Hochberg’s approach, and the standard for DEGs selection was set with *p* value < 0.01 & |Log_2_(Fold Change)| ≥ 1. The heatmap was plotted using the pheatmap package v1.0.12 [30]. The Wilcoxon test and ANOVA test were used to compare data between two and three groups and the alpha level of significance was set to 0.05 for the comparisons.

## 3. Results

### 3.1. The Alternaria and Colletotrichum Isolates from Citrus Leaves with ABS

Based on the results of fungal isolation and identification from the citrus leaves with typical symptoms of ABS, the species of *Alternaria* and *Colletotrichum* were only co-isolated from the Gonggan leaves in Zhaoqing, but not from the Hongjv leaves in Wanzhou (Table 1). From the three orchards in Zhaoqing, zero, one (14.3%), and two (33.3%) isolates of *Alternaria* and two (40%), two (28.6%), and one (16.7%) isolates of *Colletotrichum* were presented, respectively (Table 1). After adding carbendazol (10 μg/mL) to the isolation medium, four (50%), four (80%), and three (60%) isolates of *Alternaria* and zero, one (20%), and one (20%) isolates of *Colletotrichum* were presented, respectively (Table 1).

### 3.2. The Alternaria and Colletotrichum Species in the Fungal Communities of Citrus Leaves without or with ABS

After ITS amplicon sequencing, the data of fungal ASVs of *Alternaria* and *Colletotrichum* were extracted for the comparison between asymptomatic and ABS leaves. The average number of detected ASVs of *Alternaria* increased from 1.3 in asymptomatic leaves to 8.6 and 3 in ABS leaves from Zhaoqing and Wanzhou, respectively (Figure 2A). The average number of detected ASVs of *Colletotrichum* altered from 32.5 and 14.8 in asymptomatic leaves to 13.9 (*p* < 0.05) and 18 in ABS leaves from Zhaoqing and Wanzhou, respectively (Figure 2B). Accordingly, the average number of the relative abundance of *Alternaria* (%) increased from 0.1 to 9.9 and 0.7 (*p* < 0.05) from Zhaoqing and Wanzhou, respectively (Figure 2C). The average number of the relative abundance of *Colletotrichum* (%) increased from 11 and 3.3 to 13.2 and 9.7 from Zhaoqing and Wanzhou, respectively (Figure 2D). Furthermore, the composition and structure of *Alternaria* ASVs (*p* < 0.01, Figure 3A), but not *Colletotrichum* ASVs (Figure 3B), significantly altered from asymptomatic leaves to ABS leaves, which could be differentiated by the first two components of the principal co-ordinates analysis (PCoA 1 64.3% and PCoA 2 29.4%, respectively).

### 3.3. Host Response to the Infection by Alternaria and Colletotrichum Species

At 48 and 72 h after inoculation, the necrotic spots were obviously produced on citrus leaves by *Alternaria* sp. F12A and *Colletotrichum* sp. F12C, but not by the plain PDA disc (Figure 4A). The necrotic spots by F12A significantly developed, through the leaf veins, from 5 cm after 48 h to 6.1 cm after 72 h on average (*p* < 0.01). In comparison, the necrotic spots by F12C were restricted to the inoculation site, the average spot length was not significantly changed from 1 cm after 48 h to 1.1 cm after 72 h. At both time points, the necrotic spots produced by F12A were significantly longer than those produced by F12C (*p* < 0.01, Figure 4B,C).

To confirm the different virulence of *Alternaria* sp. F12A and *Colletotrichum* sp. F12C to citrus, the inoculated leaves were collected and used for transcriptional sequencing. The data showed that the citrus gene expression was strongly influenced by the infection of F12A, but was mildly influenced by the infection of F12C at both time points (Figure 5). Specifically, 5201 genes (2612 up-regulated and 2589 down-regulated genes) of 16,140 detected genes (32.2%, Figure 5A) and 6359 genes (2708 and 3651) of 15,968 detected genes (39.8%, Figure 5B) were differentially expressed genes (DEGs) after the infection of F12A for 48 and 72 h, respectively. In comparison, 1555 genes (1024 and 531) of 16,075 detected genes (9.7%, Figure 5C) and 511 genes (238 and 273) of 16,034 detected genes (3.2%, Figure 5D) were DEGs after the infection of F12C for 48 and 72 h, respectively.

### 3.4. Host Defense Genes to the Infection by Alternaria and Colletotrichum Species

All DEGs were later annotated to KEGG functional categories. From the results (Figure 6), the most DEGs were assigned to the KEGG terms of (1) plant–pathogen interaction, (2) plant hormone signal transduction, and (3) MAPK signaling pathway–plant. In specific, 285 (14.5% of detected genes), 191 (9.7%), and 132 (6.7%) DEGs were found from the three KEGG terms after the infection of *Alternaria* sp. F12A for 48 h (Figure 6A). These three KEGG terms were maintained at a highly varied level with 292 (12.3%), 188 (7.9%), and 150 (6.3%) DEGs detected after the infection of F12A for 72 h (Figure 6B). In comparison, 97 (14.9%), 64 (9.9%), and 59 (9.1%) DEGs were found from the three KEGG terms after the infection of *Colletotrichum* sp. F12C for 48 h (Figure 6C). The number of genes found from these KEGG terms were reduced to 21 (10.2%), 21 (10.2%), and 15 (7.3%) after the infection of F12C for 72 h (Figure 6D).

The expression data of the 59 plant core defense genes, involved in the three KEGG terms, were extracted and plotted in a heatmap (Figure 7). The figure shows that most of these core defense genes (57 and 52, 96.6% and 88.1%) were DEGs after the infection of F12A for 48 and 72 h, respectively (Figure 7, the first two columns). The genes, involved in pathogen-associated molecular pattern (PAMP) perception (*BAK1*, *CERK1*), reactive oxygen maintenance (*RbohD*), defense signal transduction (*CNGC1*, *CNGC2*), PAMP-triggered immunity (PTI) response (*WRKY22*, *FRK1*, *PR1*), and effector-triggered immunity (encoding disease resistance proteins), were commonly enriched (*p* < 0.01). The genes, involved in MAPK cascades, were either enriched (*MEKK1*, *MKK4*, *MPK4*, and *MPK6*) or depleted (*MKK1*). The genes, involved in salicylic acid (SA) signaling (*NPR1*, *TGA*, *SARD1*), were all enriched. The genes, involved in the negative regulation of jasmonic acid (JA) signaling (*JAZ*) and ethylene (ET) signaling (*EBF2*, but not *EBF1*), were enriched; while those involved in the positive regulation of JA signaling (*MYC2*) and ET signaling (*ERF1*) were depleted. In comparison, the expression pattern of the 59 plant core defense genes to the infection of F12C for 48 h (Figure 7, the third column) was similar, but milder to the infection of F12A. However, the expression of most of these 59 genes were recovered to normal level after the infection of F12C for 72 h (Figure 7, the forth column).

## 4. Discussion

### 4.1. Brown Spot Is Caused by Alternaria Species, but Not Colletotrichum Species

During the isolation of the causal agent from brown spot leaves, *Colletotrichum* isolates easily over-grew *Alternaria* isolates (Table 1), and the harvested *Colletotrichum* isolates were also pathogenic to citrus (Figure 4A). In addition, the saprophytic *Colletotrichum* can produce visible masses of fruit bodies on necrotic citrus organs, including the necrotic spots due to ABS [2,3]. These may be the reasons why brown spot in Zhaoqing has been a result of Anthracnose for such a long time. When added with carbendazol to inhibit *Colletotrichum* [7], the *Alternaria* species were isolated, and their virulence to citrus leaves was confirmed with a representative isolate (Figure 4A). The necrotic spot, caused by the *Colletotrichum* isolate F12C, was restricted in the wounded site, which conforms to the lifestyle of *Colletotrichum* pathogen on citrus [2,3]. In comparison, the necrotic spot, caused by the *Alternaria* isolate F12A, developed quickly along the leaf veins, which was caused by the distribution of the pathogen specific toxin [31,32]. From the results of amplicon sequencing, *Colletotrichum* species and sub-species were further conformed as common endophytes and saprophytes on citrus leaves [17,18], but their diversity and relative abundance were not increased in brown spot leaves, which implies their limited roles in the production of brown spots. In comparison, the relative abundance of *Alternaria* species was increased in Zhaoqing (*p* = 0.059) and Wanzhou (*p* = 0.038) in brown spot leaves, respectively.

Alternaria brown spot has been a well-known fungal disease on citrus in Wanzhou, Chongqing since its first identification in 2011 [7]; this disease has been a major concern and problem during the young period of citrus leaves, fruits, and twigs [4,7]. The outbreak time of brown spot in each year in Zhaoqing, Guangdong is close to that in Wanzhou, and also the disease mainly infects young leaves, fruits, and twigs [8,9,13]. However, brown spot was not commonly seen in winter when the cold weather would cause many wounds on citrus trees, thus, facilitating the infection with *Colletotrichum* species. Taken together, the causal agent of the brown spot in Zhaoqing should be assigned to *Alternaria* species, and ABS, rather than Anthracnose, and this species should be the major concern in fungicide screening and disease control.

### 4.2. Citrus Recovers Fast from Anthracnose, but Not ABS

The inoculation and transcriptional methods were combined to understand the virulence of the co-occurring *Alternaria* sp. F12A and *Colletotrichum* sp. F12C species to citrus, and in accordance, the strength of host response to the two species. Even though both infect young citrus leaves, the virulence of *Alternaria* proved to be fast and strong [4,7]. The spot by *Alternaria* expanded to nearly cover the whole leaf midrib within 48 h, and significantly altered the expression of nearly one third (32.2%) of the detected genes. In a similar study, Gai et al. (2023) found that the gene expression of citrus leaves was globally altered 12 h after inoculation with *Alternaria* pathogen; this pattern was further detected 24 and 48 h after the inoculation [4]. In addition, the expression of even more genes, up to 39.8% of the detected genes, were significantly altered 72 h after the inoculation, implying that the citrus leaf cells could not recover from the infection until the spot covered nearly the whole leaf (Figure 4A). In comparison, the spot by *Colletotrichum* did not expand out of the wounded site and only 9.7% and 3.2% of the detected genes were significantly altered 48 and 72 h after the inoculation (Figure 4). At the time point of 72 h, most affected genes by *Colletotrichum*, including many genes involved in plant defense, had recovered to normal expression level (Figure 5D and Figure 6D), which implies that the *Colletotrichum* species transforms its lifestyle from pathogen to saprophyte or endophyte, thus evading the surveillance of the plant’s immunity system [33,34].

### 4.3. Host Defense Genes Are More Sensitive to ABS Than Anthracnose Pathogen

Even though the response pattern of citrus to *Colletotrichum* sp. F12C was weaker and more easily recovered than to *Alternaria* sp. F12A, they activated several of the same sets of genes, such as genes involved in pathogen-associated molecular pattern (PAMP) perception, PAMP-induced immunity (PTI), effector-induced immunity (ETI), and plant hormone signaling [35,36,37]. For example, the expression of three genes *RbohD*, *RPS4*, and *SARD1* were significantly enriched at two time points and in response to both pathogens, these genes encode proteins for different aspects of plant defense, including cell production of reactive oxygen species, targeting of pathogen effectors, and transduction of plant systemically acquired resistance [38,39,40]. However, the expression of more genes, involved in fungal chitin perception, defense signal transduction, and pathogenesis-related protein production, were enriched after the infection with F12A, which implies that *Alternaria* sp. F12A uses chitin as a major PAMP to elicit strong citrus PTI reactions [35,36]. At the ETI level, the expression of 2.3 and 20 times more genes, involved in the production of disease resistance proteins, were enriched after the infection with F12A than with F12C, which implies F12A secretes and conveys more abundant effectors to citrus cells [39,41]. Taken together, the different virulence of the co-occurring *Alternaria* sp. F12A and *Colletotrichum* sp. F12C may be due to their different armaments of PAMPs and effectors, or the *Colletotrichum* species, as successful endophytes, may harbor specific strategies to suppress the citrus immunity system.

## 5. Conclusions

The brown spot on citrus in Zhaoqing, Guangdong is caused by *Alternaria* species, rather than *Colletotrichum* species. The co-occurring *Alternaria* and *Colletotrichum* species are differently presented in the brown spot taxonomically and functionally. The *Alternaria* sp. is much more virulent than the *Colletotrichum* sp., and can induce the typical symptoms of Alternaria brown spot. However, as we did not test the synergy between the co-occurring *Alternaria* and *Colletotrichum* species, the aggravation of the spot by the *Colletotrichum* species can not be excluded. In addition, we only worked on limited citrus leaf samples, thus, different results might be achieved if more leaves were involved, and other surface sterilization methods were selected. Our study suggests that *Alternaria* sp. and Alternaria brown spot, caused by this fungus, are the major problem in this citrus production area, thus the disease control, disease epidemiology study, and fungicidal screening should be diverted to Alternaria brown spot.

## Figures and Tables

**Figure 1 jof-09-01089-f001:**
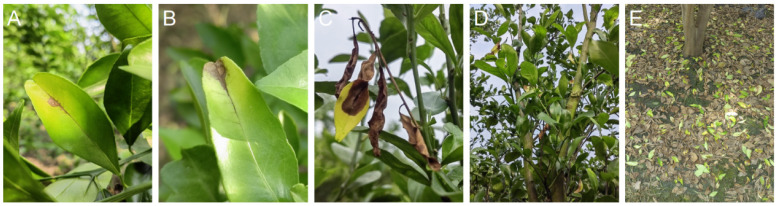
Symptoms of Alternaria brown spot (ABS) on Gonggan in Zhaoqing orchards: (**A**,**B**) necrotic spots from the middle (**A**) and the edge (**B**) of leaves with adjacent leaf veins turning black; (**C**) dead twig; (**D**,**E**) the affected leaves on the tree (**D**) and fallen on the orchard ground (**E**).

**Figure 2 jof-09-01089-f002:**
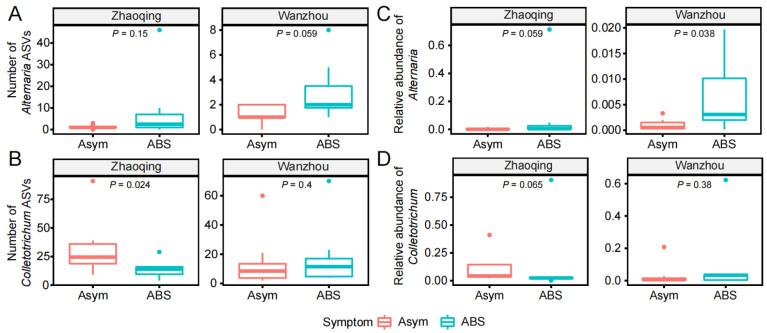
Variation in *Alternaria* and *Colletotrichum* species in leaf microbiota by Alternaria brown spot (ABS). (**A**,**B**) the number of *Alternaria* (**A**) and *Colletotrichum* (**B**) amplicon sequence variants (ASVs) between asymptomatic (Asym) and ABS leaves in Zhaoqing and Wanzhou orchards; (**C**,**D**) the relative abundances of *Alternaria* (**C**) and *Colletotrichum* (**D**) between Asym and ABS leaves in Zhaoqing and Wanzhou orchards.

**Figure 3 jof-09-01089-f003:**
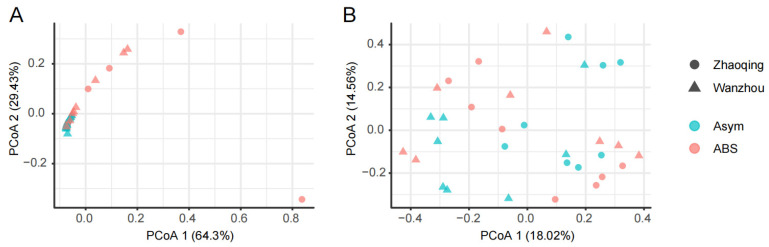
Variation in the composition and structure of *Alternaria* (**A**) and *Colletotrichum* (**B**) amplicon sequence variants in Gonggan leaves by Alternaria brown spot.

**Figure 4 jof-09-01089-f004:**
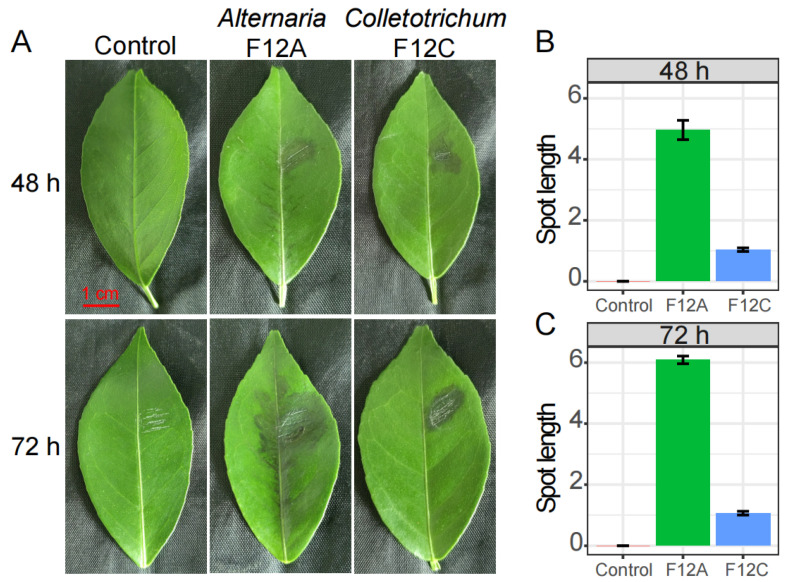
Inoculation of *Alternaria* sp. F12A and *Colletotrichum* sp. F12C on Gonggan leaves. (**A**) The development of necrotic spots 48 and 72 h after inoculation; (**B**,**C**) the length of necrotic spots along the leaf midrib 48 (**B**) and 72 h (**C**) after inoculation.

**Figure 5 jof-09-01089-f005:**
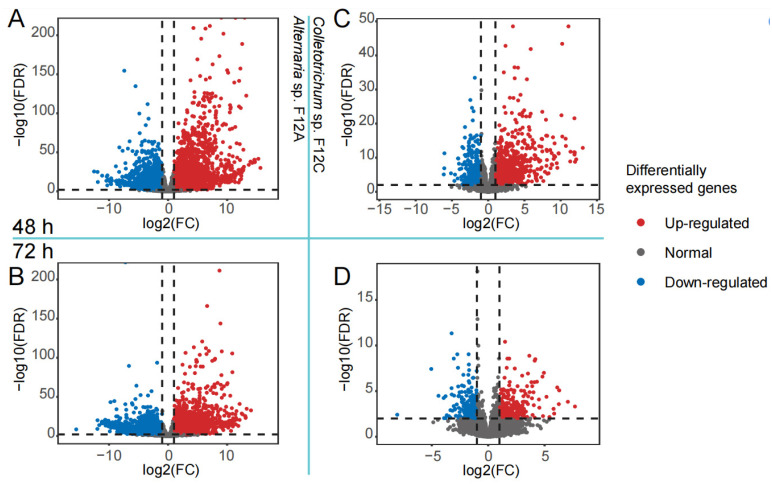
Volcano plot of differentially expressed citrus genes after the infection by *Alternaria* sp. F12A and *Colletotrichum* sp. F12C on Gonggan leaves. (**A**,**B**) Genes affected by the infection of F12A after 48 (**A**) and 72 h (**B**); (**C**,**D**) genes affected by the infection of F12C after 48 (**C**) and 72 h (**D**). FC implies the fold change of the genes expressed in infected leaves compared to control leaves, FDR implies the fdr-adjusted *p* value of the comparison.

**Figure 6 jof-09-01089-f006:**
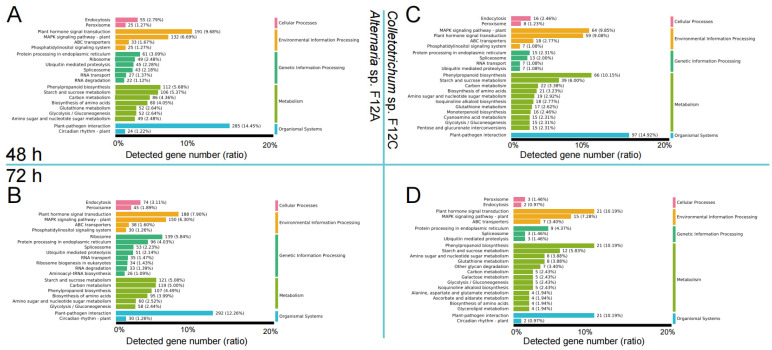
Kyoto Encyclopedia of Genes and Genomes (KEGG) categories of differentially expressed genes after the infection with *Alternaria* sp. F12A and *Colletotrichum* sp. F12C on Gonggan leaves. (**A**,**B**) Genes affected by the infection of F12A after 48 (**A**) and 72 h (**B**); (**C**,**D**) genes affected by the infection of F12C after 48 (**C**) and 72 h (**D**). The number and percentage of annotated genes are marked at the right of the bars.

**Figure 7 jof-09-01089-f007:**
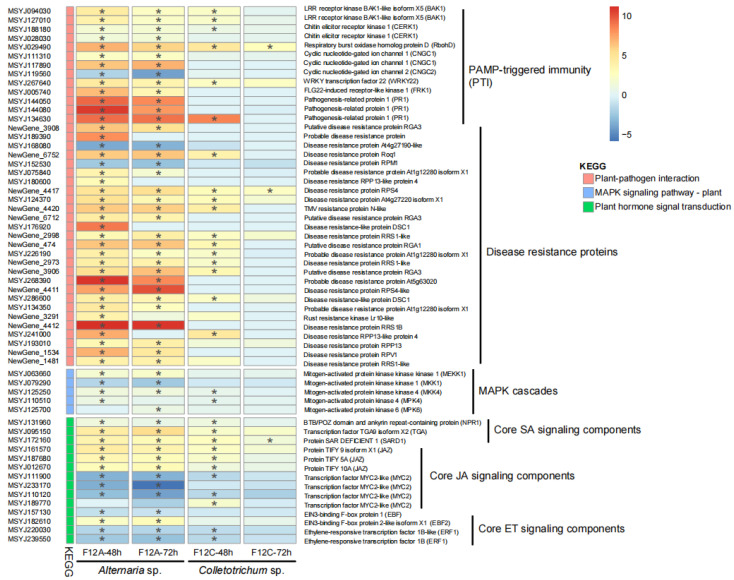
Heatmap plot of differentially expressed genes from the Kyoto Encyclopedia of Genes and Genomes (KEGG) terms of plant–pathogen interaction, Mitogen-activated protein kinases (MAPK) signaling pathway-plant, and plant hormone signal transduction 48 and 72 h after the infection with *Alternaria* sp. F12A and *Colletotrichum* sp. F12C on Gonggan leaves. The log2 fold change value is presented with grid color, and the fdr-adjusted *p* value < 0.01 is marked with “*”.

**Table 1 jof-09-01089-t001:** The information of *Alternaria* and *Colletotrichum* isolates from citrus leaves with Alternaria brown spot.

Location	Variety	Orchard	No. Segment ^a^	Carbendazol (10 μg/mL)	No. Colony	*Alternaria* spp. ^b^	*Colletotrichum* spp.
Zhaoqing, Guangdong	Gonggan	ZQ1	10	−	5	0 (0)	2 (40%)
			10	+	8	4 (50%)	0 (0)
		ZQ2	10	−	7	1 (14.3%)	2 (28.6%)
			10	+	5	4 (80%)	1 (20%)
		ZQ3	10	−	6	2 (33.3%)	1 (16.7%)
			10	+	5	3 (60%)	1 (20%)
Wanzhou, Chongqing	Hongjv	WZ1	10	−	7	0 (0)	0 (0)
			10	+	4	0 (0)	0 (0)
		WZ2	10	−	5	0 (0)	0 (0)
			10	+	4	1 (25%)	0 (0)
		WZ3	10	−	6	0 (0)	0 (0)
			10	+	4	2 (50%)	0 (0)

^a^ No., number of; ^b^ the number and ratio of *Alternaria*/*Colletotrichum* isolates.

## Data Availability

All the RNA-Seq Illumina data have been deposited at NCBI under the BioSample accession No. SAMN38043007 to SAMN38043030.

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
