# Peer review of "Comparative Study of the Co-Occurring Alternaria and Colletotrichum Species in the Production of Citrus Leaf Spot"

_jof, 2023, doi:10.3390/jof9111089_

Round 1
Reviewer 1 Report
Comments and Suggestions for Authors
This study is aimed to investigate citrus leaves with brown spot for the taxonomic and functional comparisons between the co-occurring Alternaria and Colletotrichum species. The study design is acceptable. The study contains some interesting reasults that can be considered for publication after suitable revisins.
Suggestions/comments:
Figure 3: Give in full AVS and ABS in the title.
Figure 6: letters are too small in the figures. They can not be read.
Figure 7. Give in full KEGG and MAPK in the title.
Conclusion section is missing. Please provide it.
Format of References is inconsistent. Please follow journal requirements.
Author Response
This study is aimed to investigate citrus leaves with brown spot for the taxonomic and functional comparisons between the co-occurring Alternaria and Colletotrichum species. The study design is acceptable. The study contains some interesting reasults that can be considered for publication after suitable revisins.
Suggestions/comments:
Figure 3: Give in full AVS and ABS in the title.
Response: Done as suggested. And also, it was applied to other figures.
Figure 6: letters are too small in the figures. They can not be read.
Response: Sorry for this. We will offer a vector image for Fig. 6 this time. Please check it one more time.
Figure 7. Give in full KEGG and MAPK in the title.
Response: Done as suggested.
Conclusion section is missing. Please provide it.
Response: A paragraph of Conclusion was added, please check our part 5.
Format of References is inconsistent. Please follow journal requirements.
Response: The references were revised according to the journal requirements.
Reviewer 2 Report
Comments and Suggestions for Authors
Interesting paper that focuses on a foliar leaf spot that could be caused by two different pathogens in 2 areas of China. Largest concern is that the authors never look at the possibilities of synergism between the two organisms, even though they isolated both pathogens from some lesions.
LIne 15 - delete 'Especially'; change 'entangled on the' to 'confused as causal agents of'
Line 20 - Need to better explain what was significantly altered (P<0.01).
Line 48 - change 'entanglement' to 'confusion'; change to 'for the symptoms'
Line 61 - delete 'still'
Line 62 change to '..Quangdong. Citrus...'
Line 84 - one leaf of symptomatic and one asymptomatic per tree is very minimal sampling. At least 3-5 leaves per tree would be more representative.
Line 88 - bleach treatment for 15 sec is very short for surface sterilization and is unlikely to remove epiphytic fungi very well.
Line 97 - Need to explain here in the methods why carbendazole fungicide was used. It is explained in Line 289-290, but needs to be in the methods also, especially since the reference is in Chinese.
Line 131 - Were any leaves inoculated with both pathogens? Could the fungi work synergistically instead of competing?
Line 288-290 - Does the Alternaria have a slower growth rate? Was growth rate assessed? A longer surface sterilization of 5 min in bleach would likely enhance the isolation of the Alternaria, especially if it is slow growing.
Comments on the Quality of English Language
English needs a bit of work, generally for using an incorrect word. Please see the line by line comments above that will give examples of a few places where the wording needs to be changed.
Author Response
Interesting paper that focuses on a foliar leaf spot that could be caused by two different pathogens in 2 areas of China. Largest concern is that the authors never look at the possibilities of synergism between the two organisms, even though they isolated both pathogens from some lesions.
Response: We really appreciate this comment. The suggestion is reasonable and should have been tested. However at this stage, we found it was not easy to repeat the inoculation and transcriptional sequencing as we lack of further funding on this project.
To our knowledge, Colletotrichum species has been reported as saprophytes/latent pathogens for a long time, they have not shown any additive effects to other pathogens on citrus. From our inoculation (Fig. 4), F12C only caused symptom on leaf wounds, and its caused spot did not expand to living citrus cells. This confers that F12C is saprophytic pathogen, it will not cause further damage to the leaf. In addition, the virulence of F12C was rather weaker than F12A (roughly 1/6), so we infer that F12C may have very limited synergism to F12A. Based on these, we hope that you still think our experimental design is acceptable.
LIne 15 - delete 'Especially'; change 'entangled on the' to 'confused as causal agents of'
Response: Done as suggested.
Line 20 - Need to better explain what was significantly altered (P<0.01).
Response: It referred to the composition of Alternaria, you can see it from our Fig. 3A. However, it did not have specific data to describe the variation like the average relative abundance of Alternaria increased.
Line 48 - change 'entanglement' to 'confusion'; change to 'for the symptoms'
Response: Done as suggested.
Line 61 - delete 'still'
Response: Done as suggested.
Line 62 change to '..Quangdong. Citrus...'
Response: Done as suggested.
Line 84 - one leaf of symptomatic and one asymptomatic per tree is very minimal sampling. At least 3-5 leaves per tree would be more representative.
Response: Thank you for your reminding, we should have sampled more leaves from each tree. In total, we sampled in six orchards at two locations, and collected at least 60 leaves. We hope this can be a remedy to our sampling.
Line 88 - bleach treatment for 15 sec is very short for surface sterilization and is unlikely to remove epiphytic fungi very well.
Response: The reviewer is right. We did not harvest Alternaria pathogens from the isolation in a high frequency. During our experiment, the leaves with brown spot were very young, and were not fully expanded, so we chose the bleach treatment for 15 sec. If the leaf was older, we would have extended the bleach treatment time. The short bleach treatment time might caused that many other fungi grew to contaminate the isolation.
Line 97 - Need to explain here in the methods why carbendazole fungicide was used. It is explained in Line 289-290, but needs to be in the methods also, especially since the reference is in Chinese.
Response: Down as suggested.
Line 131 - Were any leaves inoculated with both pathogens? Could the fungi work synergistically instead of competing?
Response: Thank you again for this suggestion. We responded to this suggestion following the first suggestion. The publication of this manuscript will largely help us to get further funding, then we will test this with combined method of inoculation and transcriptional sequencing. Sorry for can’t test this hypothesis this time.
Line 288-290 - Does the Alternaria have a slower growth rate? Was growth rate assessed? A longer surface sterilization of 5 min in bleach would likely enhance the isolation of the Alternaria, especially if it is slow growing.
Response: The Alternaria isolates did not grow very slow, but they can not grow over the Colletotrichum isolates. Colletotrichum are fast-growing fungi from Sordariomycetes, they are very common in and on citrus leaves across the world. The citrus leaves we chose were very young, so a sterilization for 5 min might be too long, but we believe in the reviewer, in that an extended sterilization time will improve our chance to get Alternaria.
Reviewer 3 Report
Comments and Suggestions for Authors
The manuscript was well presented. Although not really innovative it is an interesting report. However, in the Conclusion how would the new findings help to improve the control measures of the disease? In the introduction it is not clear if one or more species of Alternaria and Colletotricum were involved. If more than one it should be so stated or use "spp."
Comments on the Quality of English Languageline 49, delete "of", delete the second "first"; line 55 delete "seemed as"; line 57 rephrase "fungicide to control Colletotricum gleosporum "; line 58 rephrase as"gleosporum which widely exists with citrus" ; line 62 change "," to "." and start a new sentence with "Citrus"; line 68 delete "were"; line 82 rephrase as "varieties were"; lone 83 change "ten" to "10"; line 98 delete "the condition of";line 99 rephrase ? size of the mycelium"; change "tip" to "margin"; line 101 delete "produced"; line 102 any taxonomic identification key based on morphology was used? line 134 delete "the condition"; line 139 delete "for"; line 140, 146, 216 change "empty" to "plain"; line 142 change "in" to "at"; page 7 A title: the fungal names should be in italics; line 285 change "grown" to "grew", line 289 delete "seem as"; line 309 insert "and this species" before "should"
Round 2
Reviewer 2 Report
Comments and Suggestions for Authors
The changes addressed the wording changes. However, the other comments were not addressed in the revision. That needs to be fixed. The conclusions need to include a statement that synergism cannot be excluded and was not tested. The conclusions need to include a statement that limited samples per tree were done and that additional testing might give different results/conclusions. The conclusions also need to explain that the very short time of surface sterilization gave many cultures and that a longer time would likely have resulted in more Alternaria and less Colletotrichum.
Author Response
The changes addressed the wording changes. However, the other comments were not addressed in the revision. That needs to be fixed. The conclusions need to include a statement that synergism cannot be excluded and was not tested. The conclusions need to include a statement that limited samples per tree were done and that additional testing might give different results/conclusions. The conclusions also need to explain that the very short time of surface sterilization gave many cultures and that a longer time would likely have resulted in more Alternaria and less Colletotrichum.
Response: We express our sorry again for that we did not fulfill your suggestion, it was really a good one. Also, we appreciate that you give us another chance on this manuscript. The statements were added in the Conclusion. Please check it for detail.